# REMOVING STRUCTURED NOISE WITH DIFFUSION MODELS

## ABSTRACT

Solving ill-posed inverse problems requires careful formulation of prior beliefs over the signals of interest and an accurate description of their manifestation into noisy measurements. Handcrafted signal priors based on e.g. sparsity are increasingly replaced by data-driven deep generative models, and several groups have recently shown that state-of-the-art score-based diffusion models yield particularly strong performance and flexibility. In this paper, we show that the powerful paradigm of posterior sampling with diffusion models can be extended to include rich, structured, noise models. To that end, we propose a joint conditional reverse diffusion process with learned scores for the noise and signal-generating distribution. We demonstrate strong performance gains across various inverse problems with structured noise, outperforming competitive baselines that use normalizing flows and adversarial networks. This opens up new opportunities and relevant practical applications of diffusion modeling for inverse problems in the context of non-Gaussian measurements.

## 1 INTRODUCTION

Many signal and image processing problems, such as denoising, compressed sensing, or phase retrieval, can be formulated as inverse problems that aim to recover unknown signals from (noisy) observations. These ill-posed problems are, by definition, subject to many solutions under the given measurement model. Therefore, prior knowledge is required for a meaningful and physically plausible recovery of the original signal. Bayesian inference and maximum a posteriori (MAP) solutions incorporate both signal priors and observation likelihood models. Choosing an appropriate statistical prior is not trivial and is often dependent on both the application as well as the recovery task.

Before deep learning, sparsity in some transform domain has been the go-to prior in compressed sensing (CS) methods (Eldar & Kutyniok, 2012), such as iterative thresholding (Beck & Teboulle, 2009) or wavelet decomposition (Mallat, 1999). At present, deep generative modeling has established itself as a strong mechanism for learning such priors for inverse problem-solving. Both generative adversarial networks (GANs) (Bora et al., 2017) and normalizing flows (NFs) (Asim et al., 2020; Wei et al., 2022) have been applied as natural signal priors for inverse problems in image recovery. These data-driven methods are more powerful compared to classical methods, as they can accurately learn the natural signal manifold and do not rely on assumptions such as signal sparsity or hand-crafted basis functions. Recently, diffusion models have shown impressive results for both conditional and unconditional image generation and can be easily fitted to a target data distribution using score matching (Vincent, 2011; Song et al., 2020). These deep generative models learn the score of the data manifold and produce samples by reverting a diffusion process, guiding noise samples towards the target distribution. Diffusion models have achieved state-of-the-art performance in many downstream tasks and applications, ranging from state-of-the-art text-to-image models such as DALL-E 2 (Ramesh et al., 2022) to medical imaging (Song et al., 2021b; Jalal et al., 2021a; Chung & Ye, 2022). Furthermore, understanding of diffusion models is rapidly improving and progress in the field is extremely fast-paced (Chung et al., 2022a; Bansal et al., 2022; Daras et al., 2022a; Karras et al., 2022; Luo, 2022). The iterative nature of the sampling procedure used by diffusion models renders inference slow compared to GANs and VAEs. However, many recent efforts have shown ways to significantly improve the sampling speed by accelerating the diffusion process. Inspired by momentum methods in sampling, Daras et al. (2022b) introduces a momentum sampler for diffusion models, which leads to increased sample quality with fewer function evaluations. Chung

et al. (2022b) offers a new sampling strategy, namely Come-Closer-Diffuse-Faster (CCDF), which leverages the conditional quality of inverse problems. The reverse diffusion can be initialized from the observation instead of a sample from the base distribution, which leads to faster convergence for conditional sampling. Salimans & Ho (2021) proposes a progressive distillation method that augments the training of the diffusion models with a student-teacher model setup. In doing this, they were able to drastically reduce the number of sampling steps. Lastly, many methods aim to execute the diffusion process in a reduced space to accelerate the diffusion process. While Jing et al. (2022) restricts diffusion through projections onto subspaces, Vahdat et al. (2021) and Rombach et al. (2022) run the diffusion in the latent space.

Despite this promise, current score-based diffusion methods for inverse problems are limited to measurement models with unstructured noise. In many image processing tasks, corruptions are however highly structured and spatially correlated. Relevant examples include interference, speckle, or haze. Nevertheless, current conditional diffusion models naively assume that the noise follows some basic tractable distribution (e.g. Gaussian or Poisson). Beyond the realm of diffusion models, Whang et al. (2021) extended normalizing flow (NF)-based inference to structured noise applications. However, compared to diffusion models, NFs require specialized network architectures, which are computationally and memory expensive.

Given the promising outlook of diffusion models, we propose to learn score models for both the noise and the desired signal and perform joint inference of both quantities, coupled via the observation model. The resulting sampling scheme enables solving a wide variety of inverse problems with structured noise.

The main contributions of this work are as follows:

- We propose a novel joint conditional posterior sampling method to efficiently remove structured noise using diffusion models. Our formulation is compatible with many existing iterative sampling methods for score-based generative models.

- We show strong performance gains across various challenging inverse problems involving structured noise compared to competitive state-of-the-art methods based on NFs and GANs.

- We demonstrate improved robustness on out-of-distribution signals compared to baselines.

## 2 PROBLEM STATEMENT

Many image reconstruction tasks can be formulated as an inverse problem with the basic form

$$\mathbf{y} = \mathbf{A}\mathbf{x} + \mathbf{n}, \tag{1}$$

where $\mathbf{y} \in \mathbb{R}^m$ is the noisy observation, $\mathbf{x} \in \mathbb{R}^d$ the desired signal or image, and $\mathbf{n} \in \mathbb{R}^m$ the additive noise. The linear forward operator $\mathbf{A} \in \mathbb{R}^{m \times d}$ captures the deterministic transformation of $\mathbf{x}$. Maximum a posteriori (MAP) inference is typically used to find an optimal solution $\hat{\mathbf{x}}_{\text{MAP}}$ that maximizes posterior density $p_{X|Y}(\mathbf{x}|\mathbf{y})$:

$$\hat{\mathbf{x}}_{\text{MAP}} = \arg\max_{\mathbf{x}} \log p_{X|Y}(\mathbf{x}|\mathbf{y}) \tag{2}$$

$$= \arg\max_{\mathbf{x}} \left[ \log p_{Y|X}(\mathbf{y}|\mathbf{x}) + \log p_X(\mathbf{x}) \right], \tag{3}$$

where $p_{Y|X}(\mathbf{y}|\mathbf{x})$ is the likelihood according to the measurement model and $\log p_X(\mathbf{x})$ the signal prior.

Assumptions on the stochastic corruption process $\mathbf{n}$ are of key importance too, in particular for applications for which this process is highly structured. However, most methods assume i.i.d. Gaussian distributed noise, such that the forward model becomes $p_{Y|X}(\mathbf{y}|\mathbf{x}) \sim \mathcal{N}(\mathbf{A}\mathbf{x}, \sigma_N^2 \mathbf{I})$. This naturally leads to the following simplified problem:

$$\hat{\mathbf{x}}_{\text{MAP}} = \arg\min_{\mathbf{x}} \frac{1}{2\sigma_N^2} ||\mathbf{y} - \mathbf{A}\mathbf{x}||_2^2 - \log p_X(\mathbf{x}). \tag{4}$$

However, as mentioned, this naive assumption can be very restrictive as many noise processes are much more structured and complex. A myriad of problems can be addressed under the formulation

of equation 1, given the freedom of choice for the noise source $\mathbf{n}$. Therefore, in this work, our aim is to solve a more broad class of inverse problems defined by any arbitrary noise distribution $\mathbf{n} \sim p_N(\mathbf{n}) \neq \mathcal{N}$ and signal prior $\mathbf{x} \sim p_X(\mathbf{x})$, resulting in the following, more general, MAP estimator proposed by Whang et al. (2021):

$$\hat{\mathbf{x}}_{\text{MAP}} = \arg\max_{\mathbf{x}} \log p_N(\mathbf{y} - \mathbf{A}\mathbf{x}) - \log p_X(\mathbf{x}). \tag{5}$$

In this paper, we propose to solve this class of problems using flexible diffusion models. Furthermore, diffusion models naturally enable posterior sampling, allowing us to take advantage of the benefits thereof (Jalal et al., 2021b; Kawar et al., 2021; Daras et al., 2022a).

## 2.1 RELATED WORK

### 2.1.1 NORMALIZING FLOWS

Whang et al. (2021) propose to use normalizing flows (NFs) to model both the data and the noise distributions. Normalizing flows are a special class of likelihood-based generative models that make use of an invertible mapping $G : \mathbb{R}^d \to \mathbb{R}^d$ to transform samples from a base distribution $p_Z(\mathbf{z})$ into a more complex multimodal distribution $\mathbf{x} = G(\mathbf{z}) \sim p_X(\mathbf{x})$. The invertible nature of the mapping $G$ allows for exact density evaluation through the change of variables formula:

$$\log p_X(\mathbf{x}) = \log p_Z(\mathbf{z}) + \log |\det J_{G^{-1}}(\mathbf{x})|, \tag{6}$$

where $J$ is the Jacobian that accounts for the change in volume between densities. Since exact likelihood computation is possible through the flow direction $G^{-1}$, the parameters of the generator network can be optimized to maximize likelihood of the training data. Subsequently, the inverse task is solved using the MAP estimation in equation 5:

$$\hat{\mathbf{x}} = \arg\max_{\mathbf{x}} \left\{ \log p_{G_N}(\mathbf{y} - \mathbf{A}\mathbf{x}) + \log p_{G_X}(\mathbf{x}) \right\}, \tag{7}$$

where $G_N$ and $G_X$ are generative flow models for the noise and data respectively. Analog to that, the solution can be solved in the latent space rather than the image space as follows:

$$\hat{\mathbf{z}} = \arg\max_{\mathbf{z}} \left\{ \log p_{G_N}(\mathbf{y} - \mathbf{A}(G_X(\mathbf{z}))) + \lambda \log p_{G_X}(G_X(\mathbf{z})) \right\}. \tag{8}$$

Note that in equation 8 a smoothing parameter $\lambda$ is added to weigh the prior and likelihood terms, as was also done in Whang et al. (2021).

### 2.1.2 GENERATIVE ADVERSARIAL NETWORKS

Generative adversarial networks (GANs) are implicit generative models that can learn the data manifold in an adversarial manner (Goodfellow et al., 2020). The generative model is trained with an auxiliary discriminator network that evaluates the generator's performance in a minimax game. The generator $G(\mathbf{z}) : \mathbb{R}^l \to \mathbb{R}^d$ maps latent vectors $\mathbf{z} \in \mathbb{R}^l \sim \mathcal{N}(\mathbf{0}, \mathbf{I})$ to the data distribution of interest. The structure of the generative model can also be used in inverse problem solving (Bora et al., 2017). The objective can be derived from equation 3 and is given by:

$$\hat{\mathbf{z}} = \arg\min_{\mathbf{z}} \left\{ ||\mathbf{y} - AG_X(\mathbf{z})|| + \lambda ||z||_2^2 \right\}, \tag{9}$$

where $\lambda$ weights the importance of the prior with the measurement error. The $\ell_2$ regularization term on the latent variable is proportional to negative log-likelihood under the prior defined by $G_X$, where the subscript denotes the density that the generator is approximating. While this method does not explicitly model the noise, it remains an interesting comparison, as the generator cannot reproduce the noise found in the measurement and can only recover signals that are in the range of the generator. Therefore, due to the limited support of the learned distribution, GANs can inherently remove structured noise. However, the representation error (i.e. observation lies far from the range of the generator (Bora et al., 2017)) imposed by the structured noise comes at the cost of recovery quality.

## 2.2 BACKGROUND ON SCORE-BASED DIFFUSION MODELS

One class of deep generative models is known as diffusion models. These generative models have been introduced independently as score-based models (Song & Ermon, 2019; 2020) and denoising diffusion probabilistic modeling (DDPM) (Ho et al., 2020). In this work, we will consider the formulation introduced in Song et al. (2020), which unifies both perspectives on diffusion models by expressing diffusion as a continuous process through stochastic differential equations (SDE). Diffusion models produce samples by reversing a corruption process. In essence these models are networks trained to denoise its input. Through iteration of this process, samples can be drawn from a learned data distribution, starting from random noise.

The diffusion process of the data $\left\{\mathbf{x}_t \in \mathbb{R}^d\right\}_{t \in [0,1]}$ is characterized by a continuous sequence of Gaussian perturbations of increasing magnitude indexed by time $t \in [0,1]$. Starting from the data distribution at $t = 0$, clean images are defined by $\mathbf{x}_0 \sim p(\mathbf{x}_0) \equiv p(\mathbf{x})$. Forward diffusion can be described using an SDE as follows:

$$\mathrm{d}\mathbf{x}_t = f(t)\mathbf{x}_t\mathrm{d}t + g(t)\mathrm{d}\mathbf{w}, \tag{10}$$

where $\mathbf{w} \in \mathbb{R}^d$ is a standard Wiener process, $f(t) : [0,1] \rightarrow \mathbb{R}$ and $g(t) : [0,1] \rightarrow \mathbb{R}$ are the drift and diffusion coefficients, respectively. Moreover, these coefficients are chosen so that the resulting distribution $p_1(\mathbf{x})$ at the end of the perturbation process approximates a predefined base distribution $p_1(\mathbf{x}) \approx \pi(\mathbf{x})$. Furthermore, the transition kernel of the diffusion process is defined as $q(\mathbf{x}_t|\mathbf{x}) \sim \mathcal{N}(\mathbf{x}_t|\alpha(t)\mathbf{x}, \beta^2(t)\mathbf{I})$, where $\alpha(t)$ and $\beta(t)$ can be analytically derived from the SDE.

Naturally, we are interested in reversing the diffusion process, so that we can sample from $\mathbf{x}_0 \sim p_0(\mathbf{x}_0)$. The reverse diffusion process is also a diffusion process given by the reverse-time SDE (Anderson, 1982; Song et al., 2020):

$$\mathrm{d}\mathbf{x}_t = \left[f(t)\mathbf{x}_t - g(t)^2 \underbrace{\nabla_{\mathbf{x}_t} \log p(\mathbf{x}_t)}_{\text{score}}\right]\mathrm{d}t + g(t)\mathrm{d}\bar{\mathbf{w}}_t \tag{11}$$

where $\bar{\mathbf{w}}_t$ is the standard Wiener process in the reverse direction. The gradient of the log-likelihood of the data with respect to itself, a.k.a. the *score function*, arises from the reverse-time SDE. The score function is a gradient field pointing back to the data manifold and can intuitively be used to guide a random sample from the base distribution $\pi(\mathbf{x})$ to the desired data distribution. Given a dataset $\mathcal{X} = \left\{\mathbf{x}^{(1)}, \mathbf{x}^{(2)}, \ldots, \mathbf{x}^{(|\mathcal{X}|)}\right\} \sim p(\mathbf{x})$, scores can be estimated by training a neural network $s_\theta(\mathbf{x}_t, t)$ parameterized by weights $\theta$, with score-matching techniques such as the denoising score matching (DSM) objective (Vincent, 2011):

$$\theta^* = \arg\min_\theta \mathbb{E}_{t \sim U[0,1]} \left\{\mathbb{E}_{(\mathbf{x}, \mathbf{x}_t) \sim p(\mathbf{x})q(\mathbf{x}_t|\mathbf{x})} \left[\|s_\theta(\mathbf{x}_t, t) - \nabla_{\mathbf{x}_t} \log q(\mathbf{x}_t|\mathbf{x})\|_2^2\right]\right\}. \tag{12}$$

Given a sufficiently large dataset $\mathcal{X}$ and model capacity, DSM ensures that the score network converges to $s_\theta(\mathbf{x}_t, t) \simeq \nabla_{\mathbf{x}_t} \log p(\mathbf{x}_t)$. After training the time-dependent score model $s_\theta$, it can be used to calculate the reverse-time diffusion process and solve the trajectory using numerical samplers such as the Euler-Maruyama algorithm. Alternatively, more sophisticated samplers, such as ALD (Song & Ermon, 2019), probability flow ODE (Song et al., 2020), and Predictor-Corrector sampler (Song et al., 2020), can be used to further improve sample quality.

These iterative sampling algorithms discretize the continuous time SDE into a sequence of time steps $\{0 = t_0, t_1, \ldots, t_T = 1\}$, where a noisy sample $\hat{\mathbf{x}}_{t_i}$ is denoised to produce a sample for the next time step $\hat{\mathbf{x}}_{t_{i-1}}$. The resulting samples $\{\hat{\mathbf{x}}_{t_i}\}_{i=0}^T$ constitute an approximation of the actual diffusion process $\{\mathbf{x}_t\}_{t \in [0,1]}$.

## 3 METHOD

### 3.1 CONDITIONAL POSTERIOR SAMPLING UNDER STRUCTURED NOISE

We are interested in posterior sampling under structured noise. We recast this as a joint optimization problem with respect to the signal $\mathbf{x}$ and noise $\mathbf{n}$ given by:

$$(\mathbf{x}, \mathbf{n}) \sim p_{X,N}(\mathbf{x}, \mathbf{n}|\mathbf{y}) \propto p_{Y|X,N}(\mathbf{y}|\mathbf{x}, \mathbf{n}) \cdot p_X(\mathbf{x}) \cdot p_N(\mathbf{n}). \tag{13}$$

Solving inverse problems using diffusion models requires conditioning of the diffusion process on the observation $\mathbf{y}$, such that we can sample from the posterior $p_{X|Y}(\mathbf{x}, \mathbf{n}|\mathbf{y})$. Therefore, we construct a *joint conditional* diffusion process $\{\mathbf{x}_t, \mathbf{n}_t|\mathbf{y}\}_{t\in[0,1]}$, in turn producing a *joint conditional* reverse-time SDE:

$$\mathrm{d}(\mathbf{x}_t, \mathbf{n}_t) = \big[f(t)(\mathbf{x}_t, \mathbf{n}_t) - g(t)^2 \nabla_{\mathbf{x}_t, \mathbf{n}_t} \log p(\mathbf{x}_t, \mathbf{n}_t|\mathbf{y})\big]\mathrm{d}t + g(t)\mathrm{d}\bar{\mathbf{w}}_t. \tag{14}$$

We would like to factorize the posterior using our learned *unconditional* score model and tractable measurement model, given the joint formulation. Consequently, we construct two separate diffusion processes, defined by separate score models but entangled through the measurement model $p_{Y|X,N}(\mathbf{y}|\mathbf{x}, \mathbf{n})$. In addition to the original score model $s_\theta(\mathbf{x}, t)$, we introduce a second score model $s_\phi(\mathbf{n}_t, t) \simeq \nabla_{\mathbf{n}_t} \log p_N(\mathbf{n}_t)$, parameterized by weights $\phi$, to model the expressive noise component $\mathbf{n}$. These two score networks can be trained independently on datasets for $\mathbf{x}$ and $\mathbf{n}$, respectively, using the objective in equation 12. The gradients of the posterior with respect to $\mathbf{x}$ and $\mathbf{n}$ are now given by:

$$\nabla_{\mathbf{x}_t} \log p(\mathbf{x}_t, \mathbf{n}_t|\mathbf{y}) \simeq \nabla_{\mathbf{x}_t} \log p(\mathbf{x}_t) + \nabla_{\mathbf{x}_t} \log p(\hat{\mathbf{y}}_t|\mathbf{x}_t, \mathbf{n}_t)$$
$$\simeq s_{\theta^\star}(\mathbf{x}_t, t) + \nabla_{\mathbf{x}_t} \log p(\hat{\mathbf{y}}_t|\mathbf{x}_t, \mathbf{n}_t), \tag{15}$$

$$\nabla_{\mathbf{n}_t} \log p(\mathbf{x}_t, \mathbf{n}_t|\mathbf{y}) \simeq \nabla_{\mathbf{n}_t} \log p(\mathbf{n}_t) + \nabla_{\mathbf{n}_t} \log p(\hat{\mathbf{y}}_t|\mathbf{x}_t, \mathbf{n}_t)$$
$$\simeq s_{\phi^\star}(\mathbf{n}_t, t) + \nabla_{\mathbf{n}_t} \log p(\hat{\mathbf{y}}_t|\mathbf{x}_t, \mathbf{n}_t), \tag{16}$$

where $\hat{\mathbf{y}}_t$ is a sample from $p(\mathbf{y}_t|\mathbf{y})$, and $\{\mathbf{y}_t\}_{t\in[0,1]}$ is an additional stochastic process that essentially corrupts the observation along the SDE trajectory together with $\mathbf{x}_t$. As $p(\mathbf{y}_t|\mathbf{y})$ is tractable, we can easily compute $\hat{\mathbf{y}}_t = \alpha(t)\mathbf{y} + \beta(t)\mathbf{A}\mathbf{z}$, using the reparameterization trick with $\mathbf{z} \in \mathbb{R}^d \sim \mathcal{N}(\mathbf{0}, \mathbf{I})$, see Song et al. (2021b). Subsequently, the approximation in equation 15 and equation 16 can be substituted for the conditional score in equation 14, resulting in two entangled diffusion processes:

$$\begin{cases} \mathrm{d}\mathbf{x}_t &= \big[f(t)\mathbf{x}_t - g(t)^2 \{s_{\theta^\star}(\mathbf{x}_t, t) + \nabla_{\mathbf{x}_t} \log p(\hat{\mathbf{y}}_t|\mathbf{x}_t, \mathbf{n}_t)\}\big]\mathrm{d}t + g(t)\mathrm{d}\bar{\mathbf{w}}_{X,t} \\ \mathrm{d}\mathbf{n}_t &= \big[f(t)\mathbf{n}_t - g(t)^2 \{s_{\phi^\star}(\mathbf{n}_t, t) + \nabla_{\mathbf{n}_t} \log p(\hat{\mathbf{y}}_t|\mathbf{x}_t, \mathbf{n}_t)\}\big]\mathrm{d}t + g(t)\mathrm{d}\bar{\mathbf{w}}_{N,t} \end{cases} \tag{17}$$

which allows us to perform posterior sampling for both the signal, such that $\mathbf{x} \equiv \mathbf{x}_0 \sim p_{X|Y}(\mathbf{x}_0|\mathbf{y})$, as well as the structured noise, such that $\mathbf{n} \equiv \mathbf{n}_0 \sim p_{N|Y}(\mathbf{n}_0|\mathbf{y})$.

To solve the approximated *joint conditional* reverse-time SDE, we resort to the aforementioned iterative scheme in Section 2.2, however, now incorporating the observation via a data-consistency step. This is done by taking gradient steps that minimize the $\ell_2$ norm between the true observation and its model prediction given current estimates of $\mathbf{x}$ and $\mathbf{n}$. Ultimately, this results in solutions that are consistent with the observation $\mathbf{y}$ and have high likelihood under both prior models. The data-consistency update steps for both $\mathbf{x}$ and $\mathbf{n}$ are derived as follows:

$$\begin{aligned} \hat{\mathbf{x}}_{t-\Delta t} &= \hat{\mathbf{x}}_t - \nabla_{\mathbf{x}_t} \log p(\hat{\mathbf{y}}_t|\hat{\mathbf{x}}_t, \hat{\mathbf{n}}_t) \\ &= \hat{\mathbf{x}}_t - \nabla_{\mathbf{x}_t}||\hat{\mathbf{y}}_t - (\mathbf{A}\hat{\mathbf{x}}_t + \hat{\mathbf{n}}_t)||_2^2 \\ &= \hat{\mathbf{x}}_t - \lambda\mathbf{A}^\mathsf{T}(\mathbf{A}\hat{\mathbf{x}}_t - \hat{\mathbf{y}}_t + \hat{\mathbf{n}}_t), \quad (18) \end{aligned} \qquad \begin{aligned} \hat{\mathbf{n}}_{t-\Delta t} &= \hat{\mathbf{n}}_t - \nabla_{\mathbf{n}_t} \log p(\hat{\mathbf{y}}_t|\hat{\mathbf{x}}_t, \hat{\mathbf{n}}_t) \\ &= \hat{\mathbf{n}}_t - \nabla_{\mathbf{n}_t}||\hat{\mathbf{y}}_t - (\mathbf{A}\hat{\mathbf{x}}_t + \hat{\mathbf{n}}_t)||_2^2 \\ &= \hat{\mathbf{n}}_t - \mu(\mathbf{A}\hat{\mathbf{x}}_t - \hat{\mathbf{y}}_t + \hat{\mathbf{n}}_t), \quad (19) \end{aligned}$$

where the time difference between two steps $\Delta t = 1/T$ and $\lambda$ and $\mu$ are weighting coefficients for the signal and noise gradient steps, respectively. An example of the complete sampling algorithm is shown in Algorithm 1, which adapts the Euler-Maruyama sampler (Song et al., 2020) to jointly find the optimal data sample *and* the optimal noise sample while taking into account the measurement model in line 7 and 8 using the outcome of equation 18 and equation 19, respectively. Although we show the Euler-Maruyama method, our addition is applicable to a large family of iterative sampling methods for score-based generative models.

---

**Algorithm 1:** Joint conditional posterior sampling with Euler-Maruyama method

**Require:** $T, s_\theta, s_\phi, \lambda, \mu, \mathbf{y}$

1  $\hat{\mathbf{x}}_1 \sim \pi(\mathbf{x}), \hat{\mathbf{n}}_1 \sim \pi(\mathbf{n}), \Delta t \leftarrow \frac{1}{T}$

2

3  **for** $i = T - 1$ **to** $0$ **do**

4     $t \leftarrow \frac{i+1}{T}$

5     $\hat{\mathbf{y}}_t \sim p_{0_t}(\mathbf{y}_t | \mathbf{y})$

6

   // Data consistency steps

7     $\hat{\mathbf{x}}_{t-\Delta t} \leftarrow \hat{\mathbf{x}}_t - \lambda \mathbf{A}^\top (\mathbf{A}\hat{\mathbf{x}}_t - \hat{\mathbf{y}}_t + \hat{\mathbf{n}}_t)$

8     $\hat{\mathbf{n}}_{t-\Delta t} \leftarrow \hat{\mathbf{n}}_t - \mu(\mathbf{A}\hat{\mathbf{x}}_t - \hat{\mathbf{y}}_t + \hat{\mathbf{n}}_t)$

9  ...

9  ...

10     $\hat{\mathbf{x}}_{t-\Delta t} \leftarrow \hat{\mathbf{x}}_t - f(t)\hat{\mathbf{x}}_t \Delta t$

11     $\hat{\mathbf{x}}_{t-\Delta t} \leftarrow \hat{\mathbf{x}}_{t-\Delta t} + g(t)^2 s_\theta^*(\hat{\mathbf{x}}_t, t)\Delta t$

12     $\mathbf{z} \sim \mathcal{N}(\mathbf{0}, \mathbf{I})$

13     $\hat{\mathbf{x}}_{t-\Delta t} \leftarrow \hat{\mathbf{x}}_{t-\Delta t} + g(t)\sqrt{\Delta t}\mathbf{z}$

14

15     $\hat{\mathbf{n}}_{t-\Delta t} \leftarrow \hat{\mathbf{n}}_t - f(t)\hat{\mathbf{n}}_t \Delta t$

16     $\hat{\mathbf{n}}_{t-\Delta t} \leftarrow \hat{\mathbf{n}}_{t-\Delta t} + g(t)^2 s_\phi^*(\hat{\mathbf{n}}_t, t)\Delta t$

17     $\mathbf{z} \sim \mathcal{N}(\mathbf{0}, \mathbf{I})$

18     $\hat{\mathbf{n}}_{t-\Delta t} \leftarrow \hat{\mathbf{n}}_{t-\Delta t} + g(t)\sqrt{\Delta t}\mathbf{z}$

   **return:** $\hat{\mathbf{x}}_0$

19  **end**

---

## 3.2 TRAINING AND INFERENCE SETUP

For training the score models, we use the NCSNv2 architecture as introduced in Song & Ermon (2020) in combination with the Adam optimizer and a learning rate of $5\mathrm{e}{-}4$ until convergence. For simplicity, no exponential moving average (EMA) filter on the network weights is applied. Given two separate datasets, one for the data and one for the structured noise, two separate score models can be trained independently. This allows for easy adaptation of our method, since many existing trained score models can be reused. Only during inference, the two priors are combined through the proposed sampling procedure as described in Algorithm 1, using the adapted Euler-Maruyama sampler. We use the variance preserving (VP) SDE ($\beta_0 = 0.1, \beta_1 = 7.0$) (Song et al., 2020) to define the diffusion trajectory. During each experiment, we run the sampler for $T = 600$ iterations.

## 4 EXPERIMENTS

All models are trained on the CelebA dataset (Liu et al., 2015) and the MNIST dataset with 10000 and 27000 training samples, respectively. We downsize the images to $64 \times 64$ pixels. Due to computational constraints, we test on a randomly selected subset of 100 images. We use both the peak signal-to noise ratio (PSNR) and structural similarity index (SSIM) to evaluate our results.

### 4.1 BASELINE METHODS

The closest to our work is the flow-based noise model proposed by Whang et al. (2021), discussed in Section 2.1.1, which will serve as our main baseline. To boost the performance of this baseline and to make it more competitive we moreover replace the originally-used RealNVP (Dinh et al., 2016) with the Glow architecture (Kingma & Dhariwal, 2018). Glow is a widely used flow model highly inspired by RealNVP, with the addition of $1 \times 1$ convolutions before each coupling layer. We use the exact implementation found in Asim et al. (2020), with a flow depth of $K = 18$, and number of levels $L = 4$, which has been optimized for the same CelebA dataset used in this work and thus should provide a fair comparison with the proposed method.

Secondly, GANs as discussed in Section 2.1.2 are used as a comparison. We train a DCGAN (Radford et al., 2015), with a generator latent input dimension of $l = 100$. The generator architecture consists of 4 strided 2D transposed convolutional layers, having $4 \times 4$ kernels yielding feature maps of 512, 256, 128 and 64. Each convolutional layer is followed by a batch normalization layer and ReLU activation.

Lastly, depending on the reconstruction task, classical non-data-driven methods are used as a comparison. For denoising experiments, we use the block-matching and 3D filtering algorithm (BM3D) (Dabov et al., 2006), and in compressed sensing experiments, LASSO with wavelet basis (Tibshirani, 1996).

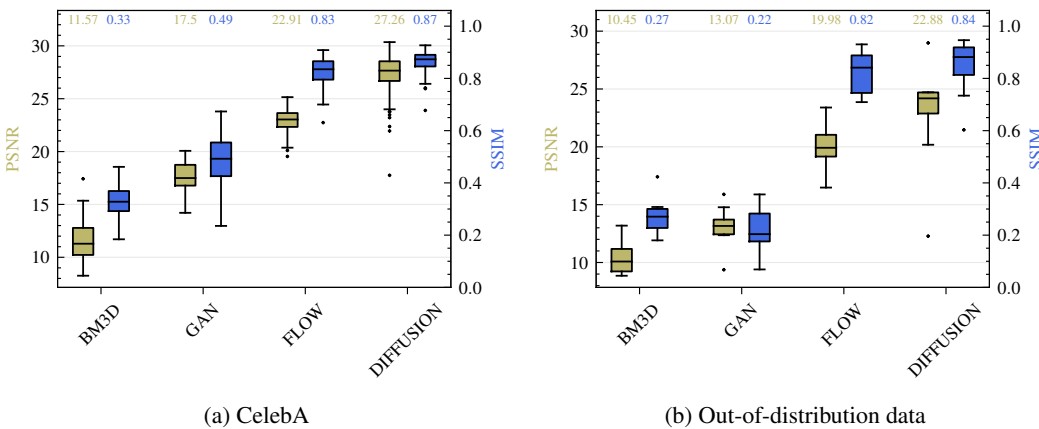

(a) CelebA  (b) Out-of-distribution data

Figure 1: Quantitative results using PSNR (green) and SSIM (blue) for the removing MNIST digits experiment on $64 \times 64$ images of the (a) CelebA and (b) out-of-distribution datasets.

Except for the flow-based method of Whang et al. (2021), none of these methods explicitly model the noise distribution. Still, they are a valuable baseline, as they demonstrate the effectiveness of incorporating a learned structured noise prior rather than relying on simple noise priors.

Automatic hyperparameter tuning for optimal inference was performed for all baseline methods on a small validation set of only 5 images. For both GAN and flow-based methods, we anneal the step size during inference based on stagnation of the objective.

## 4.2 RESULTS

### 4.2.1 REMOVING MNIST DIGITS

For comparison with Whang et al. (2021), we recreate an experiment introduced in their work, where MNIST digits are added to CelebA faces. Moreover, the experiment is easily reproducible as both CelebA and MNIST datasets are publicly available. The corruption process is defined by $\mathbf{y} = 0.5 \cdot \mathbf{x}_{\text{CelebA}} + 0.5 \cdot \mathbf{n}_{\text{MNIST}}$. In this experiment, the score network $s_\phi$ is trained on the MNIST dataset. Fig. 1a shows a quantitative comparison of our method with all baselines. Furthermore, a random selection of test samples is shown in Fig. 2 for qualitative analysis. Both our method and the flow-based method are able to recover the data, and remove most of the structured noise. However, more details are preserved using the diffusion method. In contrast, the flow-based method cannot completely remove the digits in some cases and is unable to reconstruct some subtle features present in the original images. Furthermore, we observe that for the flow-based method, initialization from the measurement is necessary to reproduce the results in Whang et al. (2021) since random initialization does not converge. The GAN method is also able to remove the digits, but cannot accurately reconstruct the faces as it is unable to project the observation onto the range of the generator. Similarly, the BM3D denoiser fails to recover the underlying signal, confirming the importance of prior knowledge of the noise in this experiment. The metrics in Fig. 1a support these observations. See Table 1 for the extended results.

Additionally, we expose the methods in a similar experiment to out-of-distribution (OoD) data. The images from this dataset not found in the CelebA dataset, which is the data used for training the models. In fact, the out-of-distribution data is generated using the stable-diffusion text-to-image model Rombach et al. (2022). We use the exact same hyperparameters as during the experiment on the CelebA dataset. Quantitative and qualitative results are shown in Fig. 1b and Fig. 3, respectively. Similarly to the findings of Whang et al. (2021); Asim et al. (2020), the flow-based method is robust to OoD data, due to their inherent invertibility. We empirically show that the diffusion method is also resistant to OoD data in inverse tasks with complex noise structures and even outperforms the flow-based methos. Unsurprisingly, the GAN method performs even more poorly when subjected to OoD data. More experiments, covering different inverse problem settings can be found in Appendix A.

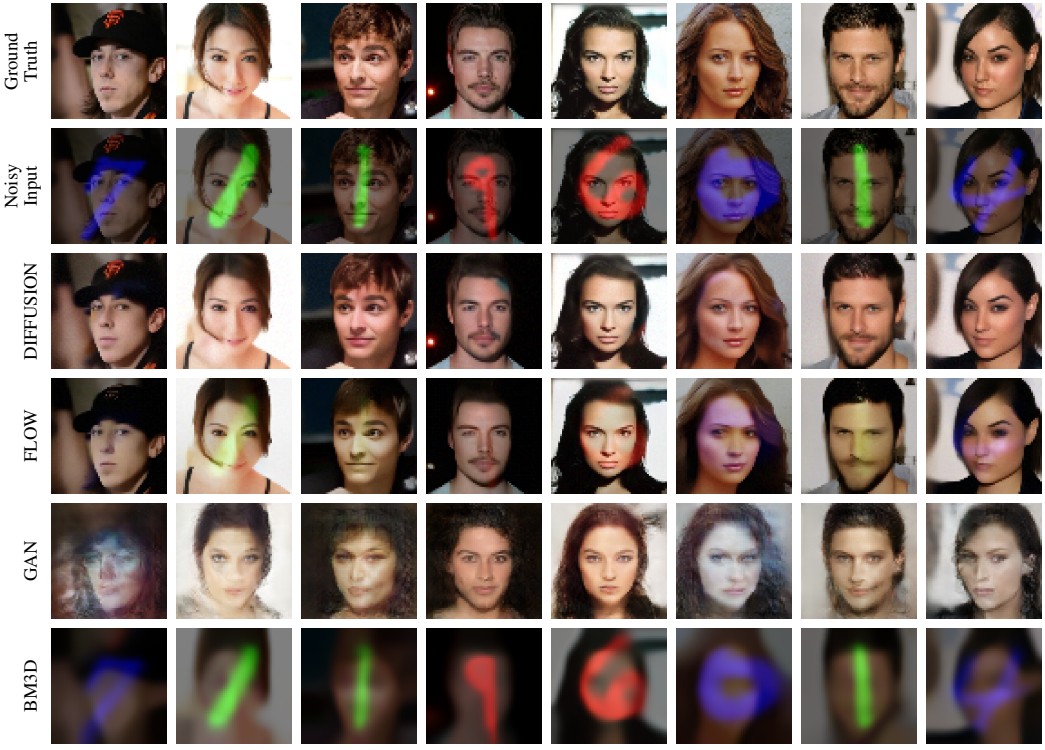

Figure 2: Results for our diffusion-based method compared to the baselines; FLOW (Whang et al., 2021), GAN (Bora et al., 2017), and BM3D (Dabov et al., 2006) on the removing MNIST digits experiment on $64 \times 64$ images of the CelebA dataset.

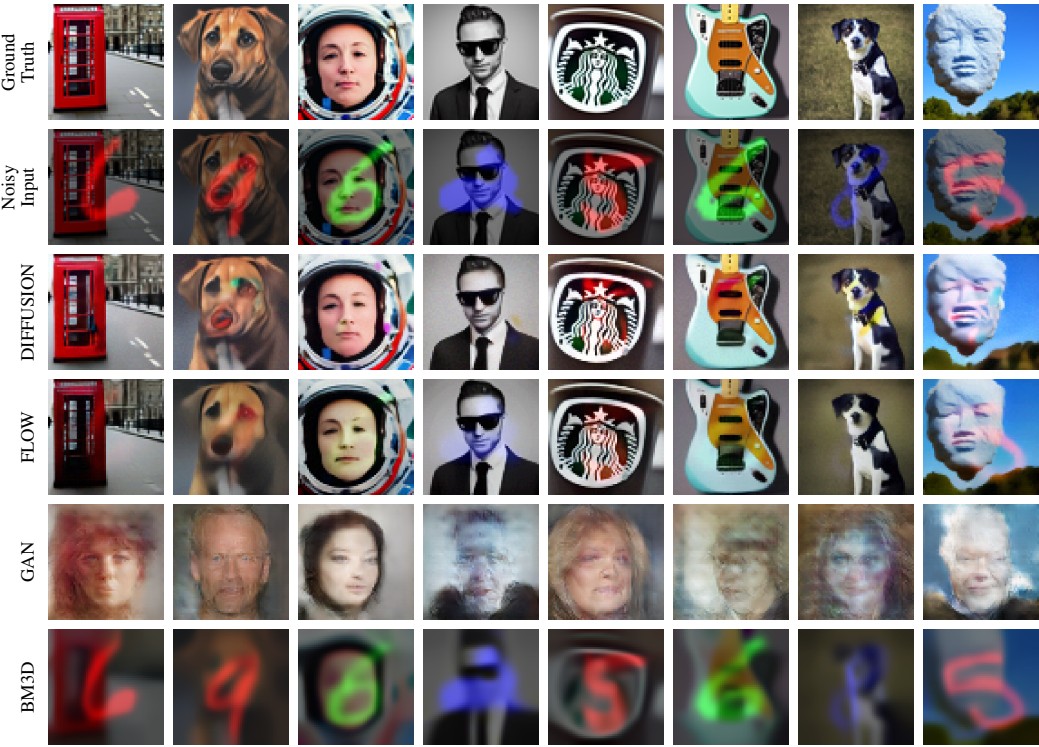

Figure 3: Results for our diffusion-based method on the removing MNIST digits experiment on an out of distribution dataset, generated using stable diffusion (Rombach et al., 2022).

### 4.2.2 PERFORMANCE

To highlight the difference in inference time between our method and the baselines, benchmarks are performed on a single 12GBytes NVIDIA GeForce RTX 3080 Ti, see Table 3 in Appendix B.2. Although this is not an extensive benchmark, a quick comparison of inference times reveals a $50\times$ difference in speed between ours and the flow-based method. All the deep generative models need approximately an equal amount of iterations ($T \approx 600$) to converge. However, for the same modeling capacity, the flow model requires a substantial higher amount of trainable parameters compared to the diffusion method. This is mainly due to the restrictive requirements imposed on the architecture to ensure tractable likelihood computation. It should be noted that no improvements to speed up the diffusion process, such as CCDF (Chung et al., 2022b) are applied for the diffusion method, giving room for even more improvement in future work.

## 5 DISCUSSION AND CONCLUSIONS

In this work, we present a framework for removing structured noise using diffusion models. Our work provides an efficient addition to existing score-based conditional sampling methods incorporating knowledge of the noise distribution. We demonstrate our method on natural and out-of-distribution data and achieve increased performance over the state-of-the-art and established conventional methods for complex inverse tasks. Additionally, the diffusion based method is substantially easier to train using the score matching objective compared to other deep generative methods and furthermore allows for posterior sampling.

While our method is considerably faster and better in removing structured noise compared to the flow-based method (Whang et al., 2021), it is not ready (yet) for real-time inference and still slow compared to GANs (Bora et al., 2017) and classical methods. Luckily, research into accelerating the diffusion process are well on their way. In addition, although a simple sampling algorithm was adapted in this work, many more sampling algorithms for score-based diffusion models exist, each of which introduces a new set of hyperparameters. For example, the predictor-corrector (PC) sampler has been shown to improve sample quality (Song et al., 2020). Future work should explore this wide increase in design space to understand limitations and possibilities of more sophisticated sampling schemes in combination with the proposed joint diffusion method. Furthermore, the range of problems to which we can apply the proposed method, can be expanded into non-linear likelihood models and extend beyond the additive noise models.

Lastly, the connection between diffusion models and continuous normalizing flows through the neural ODE formulation (Song et al., 2021a) is not investigated, but greatly of interest given the comparison with the flow-based method in this work.

## 6 REPRODUCIBILITY STATEMENT

All code used to train and evaluate the models as presented in this paper can be found at `https://anonymous.4open.science/r/iclr2023-joint-diffusion`. Essentially, the codebase in `https://github.com/yang-song/score_sde_pytorch` of Song et al. (2020) is used to train the score-based diffusion networks, for both data and structured noise, independently. To implement the proposed inference scheme, the lines in Algorithm 1 should be adapted to create a sampler that includes both trained diffusion models. Details regarding the training and inference settings used to reproduce the results in this work can be found in Section 3.2.

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

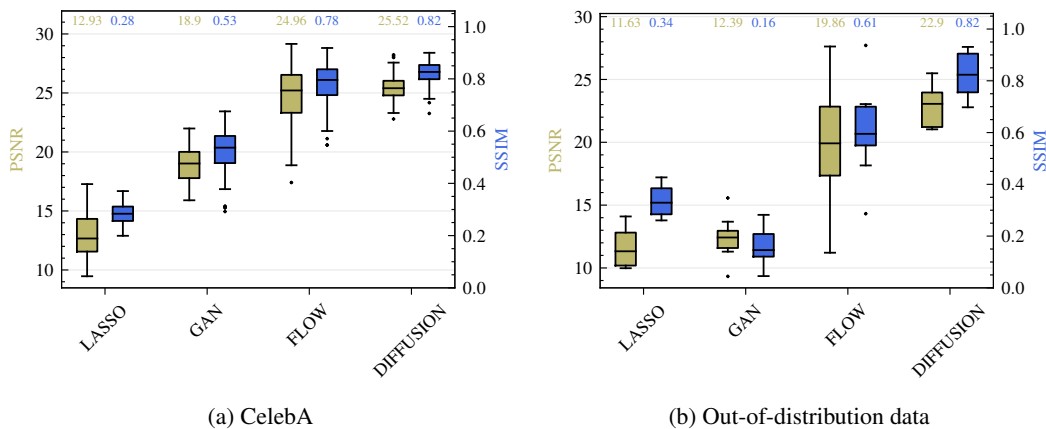

(a) CelebA                    (b) Out-of-distribution data

Figure 4: Quantitative results using PSNR (green) and SSIM (blue) for the compressed sensing with sinusoidal noise experiment on $64 \times 64$ images of the (a) CelebA and (b) out-of-distribution datasets.

## A  ADDITIONAL EXPERIMENTS

The following section explores additional inverse problems with compressed sensing and structured noise. The goal is to show the performance of the proposed method in a variety of settings.

### A.1  STRUCTURED NOISE WITH COMPRESSED SENSING

The corruption process is defined by $\mathbf{y} = \mathbf{Ax} + \mathbf{n}_{\text{sine}}$ with a random Gaussian measurement matrix $\mathbf{A} \in \mathbb{R}^{m \times d}$ and a noise with sinusoidal variance $\sigma_k \propto \exp\left(\sin\left(\frac{2\pi k}{16}\right)\right)$ for each pixel $k$. The subsampling factor is defined by the size of the measurement matrix $d/m$. In this experiment, the score network $s_\phi$ is trained on a dataset generated with sinusoidal noise samples $\mathbf{n}_{\text{sine}}$. In Fig. 5 the results of the compressed sensing experiment and the comparison with the baselines are shown for an average standard deviation of $\sigma_N = 0.2$ and subsampling of factor $d/m = 2$. Given the same hyperparameter settings, we repeat the experiment on the out-of-distribution (OoD) dataset, shown in Fig. 6. Similar to the results found in Section 4.2.1, the diffusion method is more robust to the shift in distribution and is able to deliver high quality recovery under the structured noise setting. In contrast, the flow-based method under-performs when subjected to the OoD data. Quantitative results on both CelebA and OoD are found in Fig. 4 as well as Table 1 and Table 2, respectively.

### A.2  REMOVING SINUSOIDAL NOISE

The corruption process is defined by $\mathbf{y} = \mathbf{x} + \mathbf{n}_{\text{sine}}$ where the noise variance $\sigma_k \propto \exp\left(\sin\left(\frac{2\pi k}{16}\right)\right)$ follows a sinusoidal pattern along each row of the image $k$. In this experiment, the score network $s_\phi$ is trained on a dataset generated with 1D sinusoidal noise samples $\mathbf{n}_{\text{sine}}$. See Fig. 8 for a comparison of our method to the flow-based method for varying noise variances. Both methods perform quite well, with the diffusion method having a slight edge. A visual comparison in Fig. 7, however, reveals that the diffusion method preserves more detail in general.

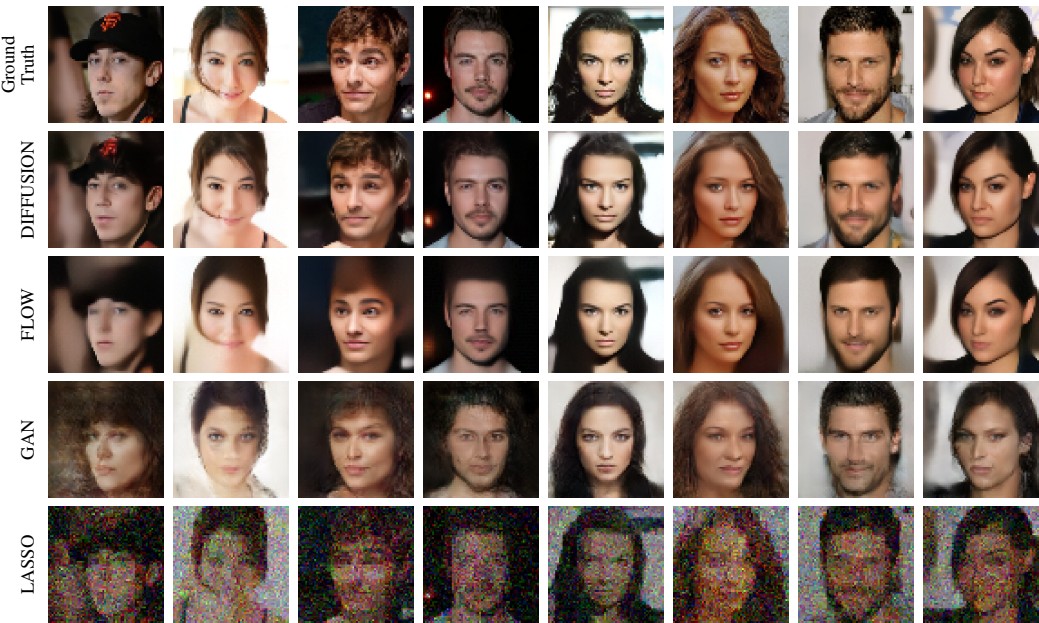

Figure 5: Comparison of results from our diffusion method compared to the baselines on the compressed sensing with sinusoidal noise experiment with $d/m = 2$, $\sigma_N = 0.2$ on $64 \times 64$ images of the CelebA dataset.

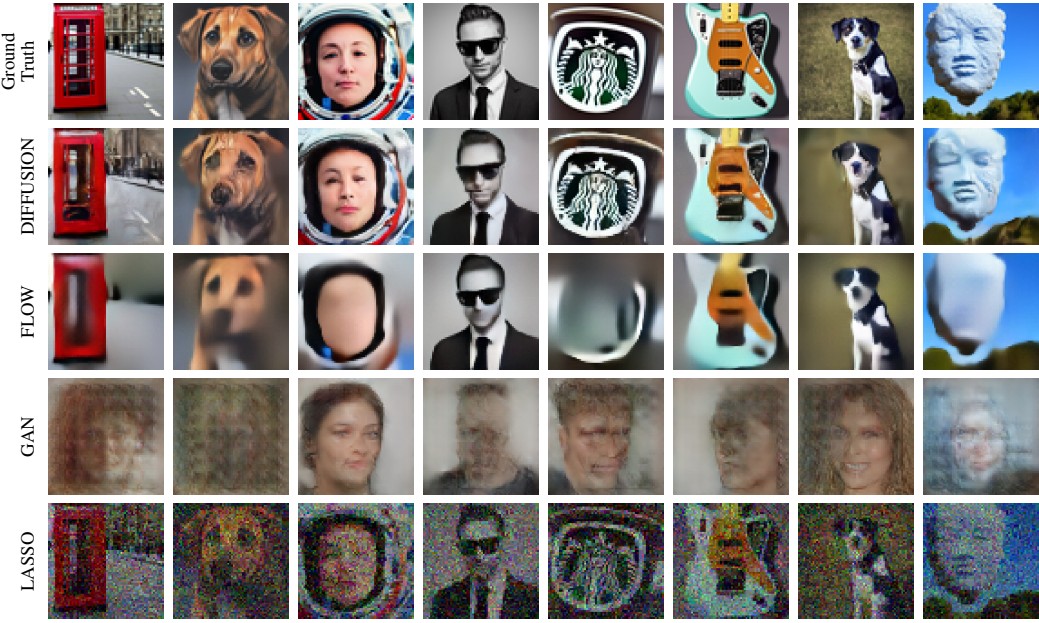

Figure 6: Results for our diffusion-based method on the compressed sensing with sinusoidal noise experiment on an out-of-distribution dataset, generated using stable diffusion (Rombach et al., 2022).

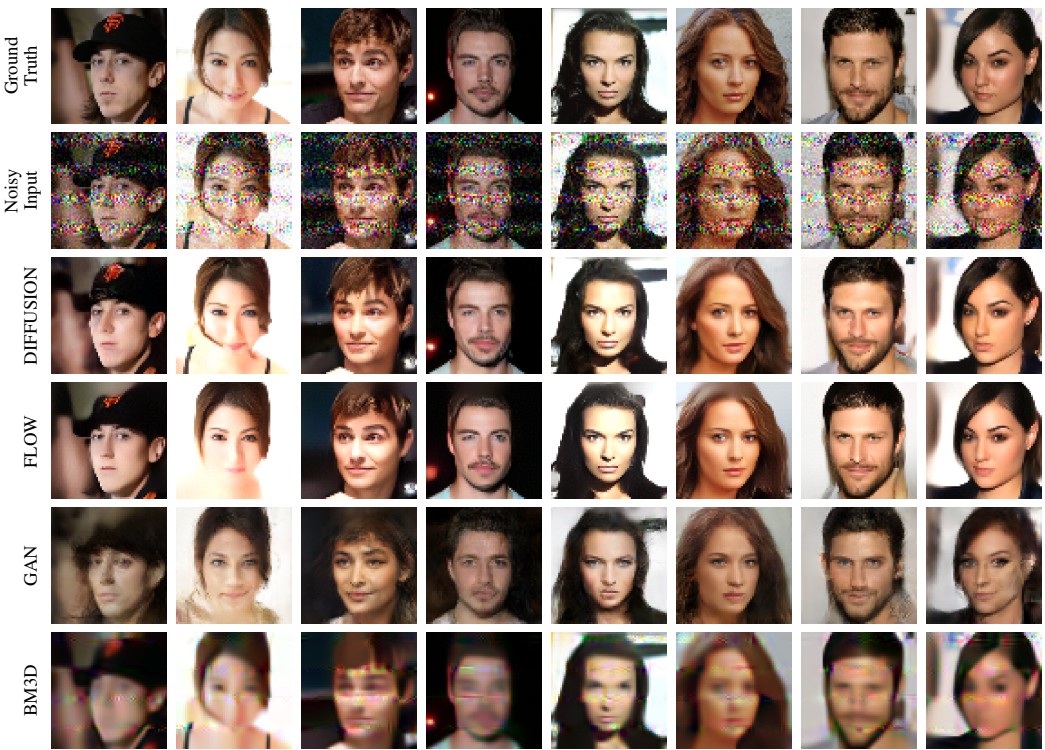

Figure 7: Comparison of results from our diffusion method compared to the baselines on the removing sinusoidal noise experiment with $\sigma_N = 0.2$ on $64 \times 64$ images of the CelebA dataset.

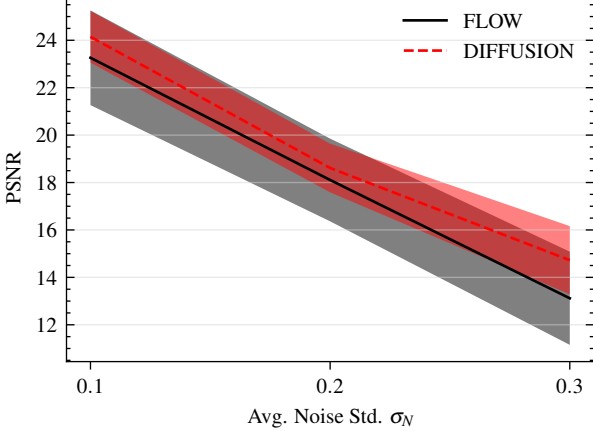

Figure 8: Comparison of PSNR values for varying sinusoidal noise variances. Shaded areas represent the standard deviation on the metric.

# B  EXTENDED RESULTS

## B.1  METRICS

Table 1: Results for the experiments and different methods on the CelebA dataset.
⋆Ours, †Whang et al. (2021), ‡Bora et al. (2017), §Dabov et al. (2006), ¶Tibshirani (1996).

|  | **MNIST** | | **CS + sine noise** | |
|---|---|---|---|---|
|  | PSNR | SSIM | PSNR | SSIM |
| ⋆DIFFUSION | 27.26 ± 1.925 | 0.865 ± 0.039 | 25.51 ± 1.040 | 0.823 ± 0.044 |
| †FLOW | 22.90 ± 1.214 | 0.827 ± 0.051 | 24.96 ± 2.292 | 0.779 ± 0.082 |
| ‡GAN | 17.50 ± 1.404 | 0.486 ± 0.099 | 18.90 ± 1.343 | 0.529 ± 0.084 |
| §BM3D | 11.56 ± 1.879 | 0.326 ± 0.059 | - | - |
| ¶LASSO | - | - | 12.93 ± 1.819 | 0.284 ± 0.037 |

Table 2: Results for the experiments and different methods on the out-of-distribution (OoD) dataset.
⋆Ours, †Whang et al. (2021), ‡Bora et al. (2017), §Dabov et al. (2006), ¶Tibshirani (1996).

|  | **MNIST** | | **CS + sine noise** | |
|---|---|---|---|---|
|  | PSNR | SSIM | PSNR | SSIM |
| ⋆DIFFUSION | 22.87 ± 4.581 | 0.842 ± 0.110 | 22.90 ± 1.568 | 0.823 ± 0.082 |
| †FLOW | 19.98 ± 1.946 | 0.824 ± 0.081 | 19.85 ± 4.840 | 0.608 ± 0.176 |
| ‡GAN | 13.06 ± 1.788 | 0.218 ± 0.088 | 12.39 ± 1.693 | 0.159 ± 0.070 |
| §BM3D | 10.44 ± 1.446 | 0.274 ± 0.069 | - | - |
| ¶LASSO | - | - | 11.62 ± 1.473 | 0.336 ± 0.057 |

## B.2  COMPUTATIONAL PERFORMANCE

Table 3: Inference performance benchmark for all methods.
⋆Ours, †Whang et al. (2021), ‡Bora et al. (2017), §Dabov et al. (2006), ¶Tibshirani (1996).

|  | Number of trainable parameters | Inference time / image [ms] | |
|---|---|---|---|
| ⋆DIFFUSION | 8.9M | 1292 | [2.153 / it] |
| †FLOW | 25.8M | 61853 | [103.1 / it] |
| ‡GAN | 3.9M | 59 | [0.059 / it] |
| §BM3D | – | 28.5 | |

## C  PSEUDO-CODE

In this section, we provide pseudo-code for the proposed joint conditional diffusion sampler with the Euler-Maruyama sampling algorithm as basis. Furthermore, we use the SDE formulation for the diffusion process which is denoted as an `sde` object with `drift`, `diffusion` and `marginal_prob` methods. The latter computes the mean and standard deviation of the diffusion transition kernel at a certain time $t$. Lastly, there are two trained score networks (NCSNv2) `score_data` and `score_noise` for the data and structured noise respectively.

```python
def joint_cond_diffusion_sampler(y, lambda_coeff, mu_coeff, num_steps):
    dt = 1/num_steps
    x = random.normal(y.shape)
    n = random.normal(y.shape)

    for t in linspace(1, 0, num_steps):
        # corrupt observation along the diffusion process
        mean, std = sde.marginal_prob(y, t)
        y_hat = mean + std * random.normal(y.shape)

        # data consistency step for x (data)
        x = x - lambda_coeff * A.T @ (A @ x - y_hat + n)

        # data consistency step for n (noise)
        n = n - mu_coeff * (n - y_hat + A @ x)

        # reverse diffusion step for x (data)
        z = random.normal(x.shape)
        x_hat = x - sde.drift(t) * x * dt
        x_hat = x_hat + sde.diffusion(t)**2 * score_data(x, t) * dt
        x_hat = x_hat + sde.diffusion(t) * sqrt(dt) * z
        x = x_hat

        # reverse diffusion step for n (noise)
        z = random.normal(n.shape)
        n_hat = n - sde.drift(t) * n * dt
        n_hat = n_hat + sde.diffusion(t)**2 * score_noise(n, t) * dt
        n_hat = n_hat + sde.diffusion(t) * sqrt(dt) * z
        n = n_hat
    # return the denoised sample x|y
    return x
```

