# OpenReview forum: "Removing Structured Noise with Diffusion Models"
_ICLR.cc/2023/Conference — Submitted to ICLR 2023_

### Official Review · Reviewer_o14W · 2022-10-22

**Confidence:** 4
**Correctness:** 2
**Technical Novelty And Significance:** 3
**Empirical Novelty And Significance:** 2
**Recommendation:** 3

**Clarity, Quality, Novelty And Reproducibility:**

Clarity & Quality: I have raised some concerns. Please see my comments above.
Novelty: The idea looks novel to me.
Reproducibility: The paper has described the reproducibility statement, but some details, such as how the score net for structured noise is trained, remain unclear.

**Strength And Weaknesses:**

Strength:

- Removing structured noise is an interesting task and the idea of the paper is fashionable.
- The experimental results look promising.

Weaknesses:

- Technical soundness & Clarify: I found the technical part of the paper hard to follow and a little bit misleading.
    * At the beginning of Section 3.1, it seems that this work tries to build a conditional distribution $p(\mathbf{x},\mathbf{n}|\mathbf{y})$ so that the inversion problem can be solved by just sampling from this distribution or taking its MAP estimation. And then to build this sampling process (or to model this distribution), a diffusion process is imposed on the "joint space" of $(\mathbf{x},\mathbf{n})$, while everything is conditioned on the noise observation $\mathbf{y}$. However, in Eq.(14), the score is conditioned on $\mathbf{y}_t$, which is further explained below in Eq.(16) as a corrupted version of $\mathbf{y}$. What is $\mathbf{y}_t$ here? Why does the conditioning depend on time $t$? What is the exact definition of $p(\mathbf{y}_t|\mathbf{y})$ and why is it related to the diffusion process of $\mathbf{x}$ and $\mathbf{n}$?
    * In Eq.(15&16), why does the second term of RHS use a sample $\hat{\mathbf{y}}_t$? Is it a good approximation?
    * In page 3 it mentioned $s_{\phi}(\mathbf{n},t)\approx\nabla_{\mathbf{n}}\log p_N(\mathbf{n})$. Is the score model $s_{\phi}(\mathbf{n},t)$ actually independent with time $t$?
    * It is not very clear how are the two score networks trained. Is $s_{\phi}(\mathbf{n},t)$ trained jointly with $s_{\theta}(\mathbf{x},t)$?
- Experiments: It is mentioned in Section 4.1 that the baseline results are improved to be more competitive. However, in Whang et al. (2021) the results seem to be better than what is reported in this paper (see Figure1&2 in Whang et al. (2021)). The inconsistency seems to make the results here questionable. Can you please provide more details on this?


Minors:
- In Eq.(13) the “proportional to” is not correct as the RHS uses log probabilities.
- In Eq.(14) the drift term should be $f(t)(\mathbf{x}_t, \mathbf{n}_t)$.

**Summary Of The Paper:**

This paper proposes to remove structured noise from noised images using a diffusion model that describes the joint distribution of the clean data samples and the structured noise. Using an additional score model for the noise, an approximate score for the joint distribution (for all time $t$) is derived. Experimental results show that the proposed method outperforms the previous benchmarks based on normalizing flows and GANs.

**Summary Of The Review:**

In conclusion, I have raised several concerns regarding the technical soundness and empirical results of the proposed method, which prevents me from giving this paper a higher rating at this time. I will consider increasing my rating if the authors address my concerns and clarify my misunderstandings.

---

> ### Author Response · Authors · 2022-11-11
> **Comment (1/2)**
>
> Thank you for the review and detailed feedback. Below we address the questions posed by the reviewer.
>
> **Q: In Eq.(14), the score is conditioned on $\mathbf{y}_t$, which is further explained below in Eq.(16) as a corrupted version of $\mathbf{y}$. What is $\mathbf{y}_t$ here?**
>
> Indeed, the joint diffusion process is conditioned on the observation $\mathbf{y}$. This was a typo and was updated in the revised manuscript (eq. 14), thank you for pointing this out. To solve this SDE, the conditional score function is factorized which leads to $\mathbf{y}_t$. Essentially, $\mathbf{y}_t$ is introduced to corrupt the observation along the diffusion process together with the intermediate generated samples $\mathbf{x}_t$. $p(\mathbf{y}_t|\mathbf{y})$ is defined exactly as a Gaussian perturbation kernel whose parameters are derived from the SDE itself. Intuitively, at each step in the reverse diffusion process, the observation is less corrupted to match the corruption of the generated samples. In the paper we provide some explanation, however, we refer the reader to [1] and [2], in which these steps are derived in more detail for a marginal conditional diffusion process.
>
> **Q: In Eq.(15\&16), why does the second term of RHS use a sample $\mathbf{\hat{y}}_t$? Is it a good approximation?**
>
> For the sake of compactness, we refer the reviewer to Appendix I.4, eq. 50 from [1] where this is justified in detail. In that section the reasonable assumptions that lead to the eq. 15 and 16 are substantiated. Another use of this term can be found in [2] Section 3.2. We also refer the reader to these specific sections in the revised paper.
>
> **Q: In page 3 it mentioned $s_\phi(\mathbf{n}, t)\approx\nabla_\mathbf{n}\log p_N(\mathbf{n})$. Is the score model  actually independent with time $t$?**
>
> This was a typo, the score model is indeed time dependent. We adopt the NCSN (noise conditional score network) network of Song \& Ermon (2020), which conditions the score network with the noise levels (which directly relate to the time step $t$ in the SDE formulation of the noise levels). In the equation you point out, it is indeed more accurate to state that the score network is not only is able to approximate the gradient of the uncorrupted data distribution, but at any point along the diffusion process, such as: $s\_{\phi}(\mathbf{n}\_t, t)\approx\nabla\_{\mathbf{n}\_t}\log p\_N(\mathbf{n}\_t)$. We will change this accordingly in the paper. Thank you for this comment.
>
> **Q: It is not very clear how are the two score networks trained. Is $s_\phi(\mathbf{n}, t)$ trained jointly with $s_\theta(\mathbf{x}, t)$?**
>
> This is a very important question, thank you for asking. There are two networks in this method, one trained on the data (CelebA), and one trained on the structured noise source. These are both trained independently, as you would do with training any generative network. In the case of the CelebA experiment with MNIST, $s_\phi(\mathbf{n}, t)$ is simply trained on the MNIST dataset, and $s_\theta(\mathbf{x}, t)$ on CelebA.
>
> We have clarified this in the following way: 1) In the method Section 3.1, we added that the two score networks for the data and structured noise can be trained independently on their respective datasets for x and n using the score matching objective. 2) In the training and inference setup Section 3.2, we added an explanation of how the two trained score networks are combined as well as added pseudo code to the Appendix. 3) We added concretely to each experiment what data was used to train the score model $s_\phi$ for the structured noise.
>
> **Q: In Whang et al. (2021) the results seem to be better than what is reported in this paper.**
>
> The reviewer points out Fig. 1 and 2 of the Whang et al paper which shows qualitative results. More representative are the quantitative metrics from both papers, which show the results for the entire test set and not a limited subset of images. When comparing the metrics shown in Fig. 3 of the Whang et al paper and Fig. 1 of our paper, we can see a similar PSNR score for the flow based method and definitely an increased score of the diffusion method. Also when cross-comparing the numbers between the different papers. More importantly, the Whang et al paper leaves out metrics on the out-of-distribution data, which shows an increased gap in performance between flow based and diffusion methods in our paper.
>
> **Q: minor comments on eq. 13 and 14.**
>
> Thank you for pointing out these minor mistakes. We have adjusted eq. 13 and 14 accordingly.

---

> ### Author Response · Authors · 2022-11-11
> **Comment (2/2)**
>
> **Additional comments**
>
> We are happy to see the reviewer acknowledges the novelty and relevance of this method. At the same time, the reviewer expresses doubts regarding the reproducibility of the work, mainly on how the score net for structured noise is trained. We hope to have given more insight into this in our comments and will clarify this in the paper. Furthermore, we will add a link to the code of this work such that reproducibility can be guaranteed.
>
> **References**
>
> 1. Song, Yang, et al. "Score-Based Generative Modeling through Stochastic Differential Equations." International Conference on Learning Representations. 2020.
> 2. Song, Yang, et al. "Solving Inverse Problems in Medical Imaging with Score-Based Generative Models." NeurIPS 2021 Workshop on Deep Learning and Inverse Problems. 2021.

---

> ### Author Response · Authors · 2022-11-17
> **Follow up comments**
>
> Dear reviewer, again many thanks for the provided feedback on our work. As the end of the discussion period is nearing, we would like to ask whether our revisions and comments have addressed any of your initial concerns about our work. Should you have additional questions, we are happy to clarify and discuss these.

---

### Official Review · Reviewer_kwKg · 2022-10-23

**Confidence:** 4
**Clarity, Quality, Novelty And Reproducibility:** The paper is clear and original.
**Correctness:** 4
**Technical Novelty And Significance:** 4
**Empirical Novelty And Significance:** 4
**Recommendation:** 8

**Strength And Weaknesses:**

Strengths:
- The paper is clearly written and well-organized.
- The paper addresses a broad class of problems. Scientists in other fields (e.g. astronomy, physics) could find this very useful.
- The method proposed is not obvious, and a significant contribution.
- The experiments are well done.

Weaknesses:
- None

**Summary Of The Paper:**

The paper proposes a new method to solve a large class of inverse problems using diffusion models. The approach is to jointly learn both a source data diffusion model and a noise diffusion model, from only a training set of noisy observations. It is assumed that the noise is added to the source signal in a known manner, but a highlight here is that the noise is allowed to be structured and not assumed to be i.i.d.

A method to jointly learn the two diffusion models is proposed. The method is then tested by using CelebA as the source signal and then adding MNIST digits as a noise model. In experiments, the diffusion model is shown to outperform alternative solutions that use GANs and and flow models.

**Summary Of The Review:**

This is exciting work, and an excellent contribution to ICLR.

---

> ### Author Response · Authors · 2022-11-11
> **Comment**
>
> Thank you for your review and comments. It is very much appreciated. We have updated the manuscript with the latest feedback.

---

### Official Review · Reviewer_uwBZ · 2022-10-24

**Confidence:** 4
**Correctness:** 3
**Technical Novelty And Significance:** 2
**Empirical Novelty And Significance:** 1
**Recommendation:** 3

**Clarity, Quality, Novelty And Reproducibility:**

Clarity: some clarity is not clear.
Quality: ordinary
Novelty: limited.
Reproducibility: not sure.


**Strength And Weaknesses:**

Strength:
1. The work additionally takes the noise structure into consideration and proposes a noise diffusion process together with an image
diffusion process to solve the inverse problem.
2. The method that samples image and noise from joint conditional posterior is rational and interpretable, and also seems easy
to implement.

Weakness
1. The right side of Eq. (14) does not contain $n_t$.
2. In algorithm1, step 7 and step 8 seem to be covered by step 10 and step 15, respectively.
3. The necessity of using the diffusion model to describe the structural noise is not convincing. Why is it better than the other generative model, such as GAN, VAE and Flow?
4. How the structural noise is embedded into the diffusion model? This point is not clear.
5. It is not clear what the training data is when training the second score function $s_{\phi}(n,t)$.
6. I wonder how the proposed method behaves compared with non-generative methods for removing MNIST digits and compressive sensing, and what is the superiority of the proposed method.


**Summary Of The Paper:**

This work proposes an approach for solving the inverse problem via diffusion models. Particularly, the work considers structured noise
that contaminates the observations. To solve the problem, another line of the diffusion process that describes noise is used except
for the image generative process. The score models in both image generation and noise generation are pretrained, and then applied
to two reverse stochastic differential equations to sample images and noises. During inference, a data-consistency step is added.
Experiments about removing MNIST digits and compressive sensing are presented to verify the effectiveness of the proposed
methods.


**Summary Of The Review:**

The work is relatively easy to follow and the idea is clear. However, the clarity to embed the structural noise is not clear. Additionally, the necessity of using the diffusion model is not convincing.

---

> ### Author Response · Authors · 2022-11-11
> **Comment**
>
> Thank you for the review and detailed feedback. Below we address the questions posed by the reviewer.
>
> **Q: The right side of Eq. (14) does not contain $\mathbf{n}_t$.**
>
> Thank you for pointing this out, we have changed eq. 14 accordingly to adjust for this typo.
>
> **Q: In algorithm1, step 7 and step 8 seem to be covered by step 10 and step 15, respectively.**
>
> In Algorithm 1, step 7 and 8 correspond to the data consistency (likelihood) terms for moving the intermediate solutions (for both $\mathbf{x}$ and $\mathbf{n}$) toward the observation. These steps originate from eq. 18 and 19. Steps 10 and 15, however, are part of the reverse diffusion process and update the intermediate solution with the drift coefficient of the SDE. These originate from solving eq. 11 (the reverse diffusion) using a numerical approach. Essentially these steps are part of moving the intermediate solutions towards their respective prior. In summary, these steps are not the same and serve their own purpose.
>
> **Q: It is not clear what the training data is when training the second score function.**
>
> Thank you for pointing this out. The second score function is trained on the structured noise source dataset. Just like we train the score function on the data itself. For example, in the CelebA with MNIST corruption experiment, the data score is learned on the CelebA dataset, and the structured noise score is learned simply on the MNIST dataset.
>
> We have clarified this in the following way: 1) In the method Section 3.1, we added that the two score networks for the data and structured noise can be trained independently on their respective datasets for x and n using the score matching objective. 2) In the training and inference setup Section 3.2, we added an explanation of how the two trained score networks are combined as well as added pseudo code in the Appendix. 3) We added concretely to each experiment what data was used to train the score model $s_\phi$ for the structured noise.
>
> **Q: How the structural noise is embedded into the diffusion model? This point is not clear.**
>
> Given a dataset comprising of noise samples, a score function can be fitted to that dataset using the score matching objective in eq. 12. We essentially have two priors, one on the noise and one on the data which we can train independently. During inference they are combined to solve the joint objective, as described in Algorithm 1. In line with the previous comment, we clarified this in the revised manuscript.
>
> **Q: I wonder how the proposed method behaves compared with non-generative methods for removing MNIST digits and compressive sensing, and what is the superiority of the proposed method.**
>
> We agree, and indeed have compared the proposed method to both deep generative methods and conventional methods designed for denoising (BM3D) and compressive sensing (LASSO) in all of the experiments. The conventional methods make crude assumptions on the statistical properties of the noise, which makes them perform poorly in the experiments with structural noise. In contrast, our method learns to approximate the true noise distribution and use this during the denoising process. This makes the proposed method superior to the conventional methods. This is also reflected in our results.
>
> **Q: Why is it better than the other generative model, such as GAN, VAE and Flow?**
>
> Diffusion models are currently considered to be the SOTA among deep generative models. Our hypothesis was that improved generative capability would also lead to better results in solving inverse problems with structured noise. This was indeed confirmed in our paper. With regard to other deep generative methods, the diffusion method is also easier to train. Diffusion models can be trained using the score-matching objective, instead of adversarial training (GAN), constraints on the architecture in order to perform exact likelihood computation (flow), or using lower bounds to estimate the true objective (VAE). Additionally, the proposed method is faster than the flow method, which is the best-performing baseline method in this paper (Appendix B). We will further highlight these differences in the paper with some additional explanations.
>
> **Additional comments**
>
> We are happy to see that the reviewer acknowledges the merits of this method. At the same time, the reviewer expresses doubts regarding the reproducibility of the work, mainly on how the structural noise is embedded using inference. We hope to have given more insight in this in our comments and will clarify this in the paper. Furthermore, we will add a link to the code of this work such that reproducibility can be guaranteed. Lastly, we hope we have been able to convince the reviewer of the advantages using the diffusion methods over the baseline methods shown in the paper.
>
> **References**
>
> 1. Jalal, Ajil, et al. "Instance-Optimal Compressed Sensing via Posterior Sampling." ICML. 2021.

---

> > ### Comment · Reviewer_uwBZ · 2022-11-21
> > **Further comments**
> >
> > Thanks for your response. I still feel confused about the following statements.
> >
> > [1] I can't entirely agree with the assumption that  "improved generative capability would also lead to better results in solving inverse problems with structured noise" since the generative model aims to generate a data sample close to the real data while the inverse problems focus on removing the noise from the data.
> >
> > [2] As shown in the main text, the modelling way for the structure noise is actually a general way to model the noise. Why do the authors emphasize the structure noise in this paper? I think the denoising effect shown in the experiment mainly comes from the training datasets. If the training dataset contains sufficient structure noise data, the model can obviously learn this denoising ability of this type of noise, which may not come from the methodology proposed in this work. Although you may argue that all the other competing methods adopt the same training dataset while obtaining worse denoising performance, it is still not convincing why the modelling way is specifically designed for the structure noise.

---

> > > ### Author Response · Authors · 2022-11-21
> > > **Replies to further comments**
> > >
> > > Thanks for engaging in the discussion, it is much appreciated. We have replied to your questions below and hope these resolve some of your concerns. If there is still anything unclear, please let us know and we are happy to further clarify.
> > >
> > > 1. It is good you point this out. Indeed, the inverse problems covered in this paper focus on removing any kind of noise from **data**. We, and many before us, argue that inverse problems can be solved by sampling from the **data** distribution, but are now conditioned on the observation, i.e. sampling from the posterior distribution. During the inverse problem, you will not only want to remove the noise from the observation as you mention but also give a sample that is clean (without the noise) **and** is a plausible solution, i.e. follows your data distribution. This is the idea with generative modeling for inverse problems, you generate a sample that is consistent with your observation through your measurement model. In the end, the generative network generates the final sample, also in this inverse setting. Therefore, improved generative capability would also lead to better samples (results). More precisely, we factorize the posterior distribution into likelihood (measurement model) and two separate generative priors, for the noise and the data. In the proposed inference scheme, both these learned distributions are used to generate the sample from the posterior distribution, and thus the quality of these generative networks is of importance for high-quality denoising.
> > >
> > >
> > > 2. Actually, explicitly modeling the noise from the data (the training dataset you mention) using deep generative models, is quite a new approach in and of itself. In fact, to the best of our knowledge, we are the first to do this using diffusion models. Our contribution provides an intuitive sampling procedure to leverage these SOTA generative models in an inverse problem setting with structured noise. Indeed as you mention, our method provides a general way to remove any kind of noise from corrupted data. However, we stress the term _structured noise_, as inverse problems with this type of noise are mainly where it benefits to use our method. Structured noise is any noise that has an arbitrarily complex probability density function and is not easily parameterized with a handcrafted function. Most methods make the faulty assumption of simple distributed noise, namely that it is Gaussian. Intuitively, solving the inverse problem fails when the noise in fact is not Gaussian distributed but has a more complex pdf, i.e. structured. Our method is able to learn the noise distribution and incorporate that knowledge into the sampling procedure, resulting in better denoising performance.
> > >
> > >
> > >     As you mention, the training dataset for training the noise prior is of importance for denoising performance. Namely, it ensures we can train an excellent prior for the noise distribution. However, without the posterior sampling algorithm proposed in this work, we cannot solve the inverse task as it is not trivial how to include the prior information we have learned for the noise. We show, we can factorize the posterior distribution and solve the inverse problem iteratively, by taking steps to both the data and the noise manifold using the respective trained score-based priors, while also enforcing data consistency through the measurement model.

---

> ### Author Response · Authors · 2022-11-17
> **Follow up comments**
>
> Dear reviewer, again many thanks for the provided feedback on our work. As the end of the discussion period is nearing, we would like to ask whether our revisions and comments have addressed any of your initial concerns about our work. Should you have additional questions, we are happy to clarify and discuss these.

---

### Official Review · Reviewer_Kqyc · 2022-10-25

**Confidence:** 5
**Correctness:** 3
**Technical Novelty And Significance:** 2
**Empirical Novelty And Significance:** 2
**Recommendation:** 5

**Clarity, Quality, Novelty And Reproducibility:**

**Clarity**
The paper is clearly written, and I had no problem understanding the motivation as well as the proposed method.

**Quality**
* Pages 2-6 are spent on background and the description of the method.  This seems a bit excessive and could be made more concise to make room for additional experiments.

**Novelty**
Given the recent success of score-based models, its application on the existing task of structured noise removal is a relatively straightforward idea.  That said, I do think a comparison between score-based prior vs. other generative priors have a merit.  This is why I hoped to see a more thorough experimental section.

**Strength And Weaknesses:**

**Strengths**
* Clear exposition and motivation.
* Problem is relevant to the field as inverse problems encompass a large family of problems and is an active area of research.

**Weaknesses**
* Experimental comparison is done on a single dataset -- especially when compared to the main baseline (Whang et al.), which considered various types of noises.
* Unclear whether the benefits are due to the difference in the model class, or other factors (e.g. hyperparameters, network size, etc).  While I don't doubt that score-based prior could outperform GAN/flow priors, it would have been nice to see some effort on trying to decouple the effects of model class vs. other architectural/implementation details.

**Summary Of The Paper:**

This paper proposes to use a score-based model for the task of removing structured noise. Specifically, the method uses a pretrained (unconditional) NCSNv2 model as the signal prior and uses modified update rule during sampling that incorporates the score of a separate noise model.  Experimental results show that score-based models achieve superior performance to existing methods based on GAN and normalizing flow.

**Summary Of The Review:**

This paper proposes to combine two score-based models -- signal prior and noise prior -- for the task of removing structured noise.  The proposed method outperforms existing deep generative priors on the task of MNIST digit removal, and while the approach is simple, this discovery has some value.  That said, the experiments and analysis are lacking.

---

> ### Author Response · Authors · 2022-11-11
> **Comment**
>
> Thank you for the review and detailed feedback. Below we address the questions posed by the reviewer.
>
> **Q: Experimental comparison is done on a single dataset -- especially when compared to the main baseline (Whang et al.), which considered various types of noises.**
>
> Thank you for pointing this out, not all results were indeed included in the main body (but in the Appendix). However, like Whang et al., we subject the proposed method to experiments on the CelebA dataset with several structural corruptions and inverse tasks. Namely, an experiment with corrupted data consisting of MNIST digits (Fig. 1/2/3), a compressed sensing experiment with sinusoidal noise (App: Fig. 4/5/6), and a denoising experiment with sinusoidal noise (App: Fig. 7/8). Additionally, both Wang et al, and our paper include experiments on out-of-distribution (OoD) data. However, only we show quantitative results on the OoD data. Would you propose moving some of the experiments from the Appendix to the main body?
>
> **Q: Unclear whether the benefits are due to the difference in the model class, or other factors.**
>
> We here took the to date best performing score-based models and the to date best performing baseline models. For instance, for the best competing baseline, the flow based model, we have adopted the GLOW architecture and hyperparameters optimized for the CelebA dataset stated in the GLOW paper. Since the architectural and implementational details of the deep generative models used in this paper have been optimized in previous work, we believe these models should provide a fair comparison for the proposed inference scheme. To clarify this further, we have updated Section 4.1.
>
> **Q: Pages 2-6 are spent on background and the description of the method.**
>
> In line with Q1, we could move some of the additional experiments to the main text, instead of a more detailed description of the method. Would you indeed recommend this?
>
> **Additional comments**
>
> We believe the paper has strong technical contributions and novelty. As the reviewer mentions, the method has value since it tackles a relevant problem that is significant for a wide class of inverse tasks. We also believe we have strong empirical results conducted through multiple experiments in line with those done by Whang et al.

---

> > ### Comment · Reviewer_Kqyc · 2022-11-17
> > **Response**
> >
> > Thank you authors for responding to my questions and concerns.  I appreciate the effort to provide additional experimental results.
> >
> > * Regarding moving more results from appendix to the main text -- I do think the paper could benefit from having more experimental results in the main text instead of appendix, but I don't feel strongly about it.  So I'll leave it up to the authors.

---

> ### Author Response · Authors · 2022-11-17
> **Follow up comments**
>
> Dear reviewer, again many thanks for the provided feedback on our work. As the end of the discussion period is nearing, we would like to ask whether our revisions and comments have addressed any of your initial concerns about our work. Should you have additional questions, we are happy to clarify and discuss these.

---

### Author Response · Authors · 2022-11-11
**A summary of updates**

We would like to thank all the reviewers for their time reading the paper and providing valuable feedback. For convenience, all changes to the paper have been marked in the updated version. With our comments, we hope to have removed some of the technical concerns and are encouraged to see the reviewers believe this work is: "relevant to the field" (reviewer Kqyc), "rational and interpretable, and also seems easy to implement" (reviewer uwBZ), "exciting work, and an excellent contribution to ICLR" (reviewer kwKg) and "an interesting task and [...] fashionable" (reviewer 014W).

**A. Improving reproducibility**

One of the concerns of reviewers uwBZ and 014W with the paper was reproducibility. We believe that this method is absolutely reproducible. Therefore, we have added further explanation on how to train the structured noise score model, which is straightforward once you have a noise dataset. Furthermore, we have shared our implementation (https://anonymous.4open.science/r/iclr2023-joint-diffusion) and added that to the reproducibility statement. Lastly, we included pseudo-code of the proposed joint conditional diffusion sampler in the Appendix.

**B. Slight changes to equations**

As was pointed out by reviewers uwBZ and 014W, equations 13 and 14 contained slight inaccuracies, as well as the noise score model definition on page 5, which should be dependent on $t$. This has been adjusted in the updated version.

**C. Additional experiments**

Per suggestion of reviewer Kqyc, some of the additional experiments in the Appendix can be moved over to the main text. To make room for this adjustment, part of description of the (baseline) methods should be moved to the Appendix or removed.

---

### Decision · Program_Chairs · 2023-01-20

**Decision:**

Reject

**Justification For Why Not Higher Score:**

As written in the meta-review, the experiments are very simplistic and do not cover practically relevant cases of structured noise removal.

**Justification For Why Not Lower Score:**

N/A

**Metareview: Summary, Strengths And Weaknesses:**

The problem considered is relevant, and the proposed method of removing structured noise with diffusion models is intuitive and shown to perform well for removing handwritten digits from celeb-A face images. Conceptually, the proposed methods is similar to Wang et al. 2021''s approach in that it uses generative priors to remove structured noise. A major difference is that the paper under review uses score based models and proposes a corresponding conditional sampling approach, while the prior works rely on other kinds of generative models. Using score based models in this context is very reasonable since score based models perform very well amongst methods imposing generative priors.

However, the paper's experiments are very simplistic and the main experiments pertain to removing handwritten digits from celeb-A pictures. It would be more interesting to study practically relevant cases of removing structured noise from scientific data for example, where structured noise actually occurs in practice, and to compare to a wider array of methods. For example, for such scientific data, structured noise removal with networks trained end-to-end also work very well.


**Summary Of Ac-Reviewer Meeting:**

Here is a summary of the reviews and the discussion during the rebuttal period:

R1 (5) notes that the problem is relevant and the exposition is clear, but finds that experiments are lacking, in particular, experimental comparisons are done on a single dataset. The authors note that additional experiments are in the appendix, however, those experiments do not go significantly beyond those in the main body.

R2 (3) notes that the method is easy to implement and interpretable, and that the exposition is clear. However, R2 notes that it is unclear how the method compares to traditional approaches for removing structured noise and about the comparative strength of the method and asks for a variety of clarifications. The authors provided those clarifications and note that the paper compared to classical approaches like BM3D and LASSO.

However, even the comparisons to lasso (as I understand them from the paper) are not entirely fair since lasso doesn't use a structured noise model in reconstruction, even though it could (by representing the noise in another sparse basis than the signal). Similarly, the comparison to BM3D is not fair in that BM3D is designed for Gaussian noise removal and is known to not work well for structured noise. Those methods are set up to fail in the studies scenario, but in more realistic structure noise removal setups, well-performing alternative approaches are available.

R3 (8): find that the paper addresses a broad class of problems and that the method proposed is not obvious, and finds that the experiments are well done. The reviewer identifies no weaknesses.

R4 (3) finds the technical part of the paper hard to follow and gives concrete examples, and notes that the baseline results in the related paper Wang et al. 2021 seem to be better than in the paper under review. The authors respond that they obtain similar scores as the Wang et al. paper.